# Māori Experiences and Beliefs about Antibiotics and Antimicrobial Resistance for Acute Upper Respiratory Tract Symptoms: A Qualitative Study

**DOI:** 10.3390/antibiotics11060714

**Published:** 2022-05-26

**Authors:** Kayla Hika, Matire Harwood, Stephen Ritchie, Amy Hai Yan Chan

**Affiliations:** 1School of Medicine, Faculty of Medical and Health Sciences, University of Auckland, Auckland 1142, New Zealand; khik663@aucklanduni.ac.nz; 2General Practice and Primary Healthcare, School of Population Health, Faculty of Medical and Health Sciences, University of Auckland, Auckland 1142, New Zealand; m.harwood@auckland.ac.nz; 3Molecular Medicine and Pathology, School of Medical Science, Faculty of Medical and Health Sciences, University of Auckland, Auckland 1142, New Zealand; s.ritchie@auckland.ac.nz; 4Auckland District Health Board, Auckland 1023, New Zealand; 5School of Pharmacy, Faculty of Medical and Health Sciences, University of Auckland, Auckland 1142, New Zealand

**Keywords:** Māori, antibiotics, beliefs, prescribing, use experiences, qualitative, antimicrobial resistance

## Abstract

Antimicrobial resistance (AMR) is a threat to public health. Addressing unnecessary antibiotic use provides an opportunity to reduce antibiotic consumption and to slow AMR. Understanding people’s beliefs is important for informing antimicrobial stewardship (AMS) initiatives. Within New Zealand, health inequities exist between Māori and non-Māori; however, no research has examined Māori beliefs about antibiotics and AMR. The aim of this study was to explore the experiences related to antibiotic use of Māori in New Zealand. In-depth, semi-structured interviews were conducted with 30 Māori adults recruited from primary care to explore the experiences, perceptions and beliefs that Māori have about antibiotics, and about AMR. Overall, 30 Māori adults (23% male; age range from 20 to 77 years) participated. Three themes emerged: systemic-, social-, and individual-related factors. From these themes, seven subthemes explained the factors that influenced antibiotic use and their perceptions of AMR in Māori: general practitioner (GP) times and ratios, effect of colonisation, lack of knowledge and information, access and poverty barriers, relationship with health professionals, illness perceptions, treatment beliefs and Whaakaro (thoughts), and beliefs pertaining to natural (rongoā) and Western medicine. Participants identified potential solutions to improve antibiotic use such as cultural support and involving Te Ao Māori; recognising these can inform future AMS initiatives.

## 1. Introduction

Antibiotic resistance has become a threat to public health at individual, community and national levels [1]. In New Zealand (NZ), the rate of infectious disease hospital admissions increased by 17,000 per annum between 1989 and 2008 [2], with marked differences in these rates by ethnicity. The rise in infection-related hospital admissions were most marked for Māori, the Indigenous peoples of New Zealand, who experience health inequities related to ethnicity including higher rates of incidence, hospitalisation and deaths [3]. Between 2004 and 2008, the age-standardised rate ratio of hospital admission for infectious disease was 2.15 times higher for Māori than for other ethnicities [4]. Not surprisingly, antibiotic dispensing for Māori is higher than for other ethnicities in NZ [4], which on the one hand reflects the higher infection burden, but on the other may be inappropriate prescribing which is potentially harmful for Māori [4,5,6]. The reasons for these inequities between Māori and non-Māori are complex and multifactorial. The inequities are driven in part by the vast differences in socioeconomic status as the opportunities provided by income, education and housing are not evenly distributed in NZ [7,8]. Due to the effects of colonisation, non-Māori have higher median incomes and educational achievement than Māori [9]. This, in addition to institutional or structural racism, has led to adverse effects on healthcare and medicine access and consequently health outcomes [7,10]. 

High rates of antibiotic dispensing for Māori, in large part driven by higher rates of infectious disease, contribute to the development of antibiotic resistance. Addressing unnecessary antibiotic use provides an important opportunity to reduce antibiotic consumption [5,11]. One example of unnecessary antibiotic prescribing is for acute upper respiratory tract symptoms (or ‘cold and flu’ symptoms). Studies have reported that whilst many general practitioners were aware that prescribing of antibiotics for cold and flu symptoms is unnecessary [12], a key driver of this unnecessary antibiotic prescribing was either to meet patients’ expectations for antibiotics, or the general practitioner’s perceptions of their patients’ behaviour and expectations [13,14]. Key drivers of these expectations are the patients’ beliefs about antibiotics and their general perceptions of healthcare [15]. 

Understanding people’s beliefs and perceptions about antibiotics is important for informing antimicrobial stewardship initiatives, as perceptions can have a significant influence on the patient-health professional interaction and can impact antibiotic use [16,17]. General practitioners have suggested that interventions that target the public and their perceptions of antibiotics would be useful to reduce unnecessary prescribing [12]. Within NZ, a study of Samoan people’s perceptions, beliefs and knowledge of antibiotics found that more than two-thirds of participants thought that antibiotics were for pain relief and most (81%) perceived antibiotics as a useful treatment for common colds/flu. Interestingly, the participants held a more traditional understanding of health, rather than a Western view [16]. The study concluded that health professionals should not assume patients hold the same Western understanding of infection and antibiotics [16], and that understanding a patient’s world view and perceptions is an important first step towards optimising medicine use. 

Differing worldviews on health can influence treatment and healthcare expectations in individuals, and health professionals’ perceptions and behaviours are influenced by these expectations, potentially contributing to suboptimal or unnecessary antibiotic prescribing [13,15]. A significant gap in current knowledge is understanding the views of antibiotic use for Indigenous Māori in NZ. Attaining such information fulfils New Zealand’s Te Tiriti o Waitangi (Treaty of Waitangi; see Appendix A for Te Reo definitions) responsibilities, has the potential to advance Māori health and achieve health equity [18]. As there have been no prior studies that have explored the perceptions of Māori towards antibiotics and antimicrobial resistance, there is a need to investigate the experiences of Māori using a qualitative research approach. Qualitative research provides rich descriptive accounts of people’s experiences to understand participants’ own worldviews rather than counting occurrences, or reducing phenomena to numerical values to carry out statistical analyses [19]. Qualitative approaches are particularly valuable when used to inform later quantitative research, for example, to inform the design of surveys or questionnaires, or to help explain quantitative data [20].

This paper thus aims to explore the experiences, perceptions and beliefs that Māori have about antibiotics and the use of antibiotics in regard to acute upper respiratory tract symptoms, and of antimicrobial resistance, using a qualitative research approach. 

## 2. Results

### 2.1. Participant Characteristics

A total of 30 participants participated in the study; 21 participants were interviewed one-on-one and the remainder were interviewed in four focus groups with one group of three and three groups of two using a semi-structured interview guide (Appendix A). Twenty-three participants were female and seven were male; their ages ranged from 20 to 77 years. A total of 28 participants consented to audio recording during the interview. Two participants did not consent to audio recording, but allowed handwritten notes to be taken.

Nine participants reported living with a long-term illness, which included breast cancer, heart disease, rheumatoid arthritis and diabetes (Appendix A). Eleven participants reported taking regular medication for their long-term conditions such as allopurinol, insulin, tramadol and codeine—none of which were identified to be for infections. Ten participants chose not to provide demographic information. 

### 2.2. Key Themes

Three overall themes influenced the use of antibiotics; these themes related to Systemic, Social and Individual factors. From these overall themes, seven subthemes emerged to explain the factors that influenced antibiotic use and resistance. Additionally, one specific theme related to potential solutions was raised by participants to improve antibiotic use (Figure 1).

#### 2.2.1. Theme 1: Systemic Factors Influencing Antibiotic Use

##### General Practitioner (GP) Times and Ratios

Most of the participants were living with chronic health conditions and, due to limited consultation time with the GP, additional problems including infections were often not able to be addressed in the consultation. This was recognised as a potential factor that could lead to suboptimal use of antibiotics in Māori, “…because the thing today is, you’re only allowed 15 minutes to try and get everything in, that was another thing with the (GP). Oh, what was it, I think it was my hearing, and something else, and when I went in there to see him and I tried [to] ask him about my hearing and about this other thing and he said ‘well, it’s one or the other,’”—P23, F, 66 years old.

Participants also reported that there was not enough consultation time with the GP to give a full narrative of all health symptoms that they were experiencing, thus impacting their understanding of their health and the role of medication and treatment, as “there’s not enough [time], I guess, when the time is confined to 15 minutes. You can’t put everything into 15 minutes, yeah,”—P9, F, 69 years old. Along with time, participants identified lack of GP availability as a contributing factor to why they could not access the care they need. One participant said that “…you’ve got to be quick to get that spot to see a doctor. There [are] not enough doctors here,”—P5, F, age unknown. Meanwhile, another highlighted the mismatch between GP availability and need for help: “…because when I need to see a doctor, she’s not available or she’s somewhere else,”—P9, F, 63 years old.

##### Effects of Colonisation—An Underlying Contributor to Misuse and Underuse of Antibiotics

The negative impacts of colonisation for Māori were both explicitly and implicitly discussed by participants. Many recognized the ‘downstream’ effects of colonisation because of how colonisation underpins the systems and structures in society that participants perceived provided advantage to NZ European ways over Māori values. Consequently, participants described experiencing disadvantaged socioeconomic and health status, which in turn undermines socioeconomic opportunities and resources such as health literacy, eventually resulting in increased risk for infections and subsequently, antibiotic overuse.

Many participants reported that the removal of traditional Rongoā (healing) knowledge had an impact on their healthcare utilisation and perceptions of medicine, as “…our Māori-ism [sic] needs to be brought into our thing.I don’t know why people don’t know anything about it, you know, like medicine for the kids, the cream that they get. Yeah it’s good, but our medicineis just… better,”—P19, M, age unknown.

Participants also reported the effects of urbanization of Māori communities, as a result of land-loss, resulted in a loss of culture and identity over time: “I was bought up on the marae… It was a good life. My mother spoke fluent Māori, more so my mum then my dad… The government at that time change(d); there was a generational change, so their house was built …so we moved from the marae up there,”—P7, F, age unknown. This culture shift impacted their health-seeking behaviour and how they integrated into the health and societal systems.

#### 2.2.2. Theme 2: Social Factors

##### Lack of Knowledge and Information concerning Antibiotics for Patients

Commonly, participants were not able to distinguish the difference between a bacterial and viral infection. Participants said they did not understand what an antibiotic was before an explanation was presented by their health provider and some reported having antibiotics for viral infections. Participants reported having had limited or unmemorable past exposure to information regarding antibiotics or that they “…can’t remember. It’s been a while. I heard [information about antibiotics] before, but I can’t remember,”—P22, M, 63 years old.

Information regarding medications such as antibiotics was perceived to be scarce. Participants reported having fears of taking antibiotics due to the lack of understanding and information regarding what they are and what they do to the body: “I get scared if I don’t know the medication. See but don’t even explain that what it is. It’s just an antibiotic, what is that, what is the medication doing for you, so I’m just more like Augmentin. I know what, I don’t know exactly what Augmentin is, but I know what it does for me. It kills what needs to be, the bacteria inside myself,”—P11, M, 46 years old. 

In terms of information sources, only a few participants reported receiving information about antibiotics from their GP. Similar to the concerns reported in Theme 1, this may be due to issues of access to the GP, and time availability during the consultation. Furthermore, some participants reported that they were not able to understand information provided by their GP, as “they do write in medical language and not everyone is medical and then you ask the pharmacy and they’re only reading off what the bottle says or on the paper,”—P16, F, age unknown. Many participants reported using the internet or friends and whānau as sources of information about antibiotics: “I ask for the name of [the medicine] and I googled [sic],”—P27, F, 44 years old.

Many participants indicated a lack of knowledge in terms of when to use antibiotics and what the purpose of the medication is for, stating that antibiotic use is required for more serious illness, for example when “[the illness is] really bad. That’s the only time I will take an antibiotic,”—P23, F, 66 years old. Participants described the ‘severity’ of how bad the infection is based on their experience of symptoms: “I guess it depends really on how mucus-y [sic] you are or whoever is,”—P5, F, age unknown.

##### Access to Healthcare and Poverty Barriers to Appropriate Antibiotic Use

Poverty and other barriers to accessing timely and quality healthcare influenced medication provision and use for the participants. Nearly all participants described having trouble accessing GP appointments due to cost, which also led to barriers in accessing antibiotics for the correct use. The cost of an appointment was highlighted as a concern for most of the participants. Many did not have the finances to seek healthcare (NZD$17 for adults) or the antibiotics (NZD$5 per medicine) when these were prescribed, as “…it costs money and it still gotta [sic] come back again for thrush, so it’s a waste of time coming here to get you [antibiotics] if I have to come back and get you [thrush medication],”—P6 —F, age unknown. Due to the cost of a GP appointment, many participants commented on the incurred debt that sat at the clinic. This debt caused them to worry that they may be turned away from necessary care and produced emotional strain on participants: “I did not want to come in this morning. I had a bit of an argument with my partner. I was telling her that, you know, that I might get here and they might say ‘no,’ because of the money I owed them… So yeah, that’s all I was afraid of coming in,”—P1, M, age unknown.

Poverty forced participants to prioritise expenditure on items other than their healthcare, which delayed access to timely care; for example, one participant had to weigh up the options of spending money on food or medication: “I’ll get them [medication], and it was like $2 or $5 and sometimes I didn’t have the finance, and when I did have the finance it was $5 food or $5 tablet,”—P26, F, 45 years old. Several participants reported the issues they faced when suffering from poverty, in which the cost of healthcare in general prohibited them from accessing timely care, as there were other expenses they needed to pay: “I’m asking for a food grant. I had to pay extra in parking. I get $4 after everything is paid and that’s what comes from work and income and that’s all we get and there are four little girls and I’ve got two teenagers at home as well. But after the rent is paid, the power is paid, that’s it, $4. Then the daycare bill, school fees, it’s like “oh please give me a break,”—P2, F, 39 years old.

Travel was also a barrier for participants to get necessary and timely healthcare. Some participants relied on whānau to get to the GP clinic while others used other modes of transport such as buses, which were associated with additional costs or time-delays. Others reported having to travel long distances to get the help they required to treat their symptoms: “that’s a half an hour, three-quarters of an hour drive [to go to the doctor]… sitting there in the car holding yourself up,”—P19, M, age unknown.

One participant highlighted a cycle of poverty and crime. The participant reported that due to poverty and debt, they had to find the financial assistance to pay for their medication by other means, such as crime and drugs, to be able to access healthcare: “…that’s usually what the crime is coming from you know. They can’t get the medicine, so they get the other drug and they’re in debt, so they have to do crime to pay it back and you know, it’s a cycle,”—P1, M, age unknown.

##### Relationship with Health Professionals

Many participants reported having a good relationship and rapport with their GP and that they adhered to the advice of their GP. In certain cases, this supported access to necessary medication in general and information, with one participant describing feeling culturally safe to share information during their consultation: “I just think they’re very culturally safe as [I] feel [they] welcome me and my whānau,”—P28, F, 29 years old.

Many also reported they took the advice of their GP, because the doctor is the health professional and was more knowledgeable about health in general than the participants themselves, as the GP “seems to be really good, I try to be honest with them ‘cause [sic] at the end of the day they know more than I know, and I need their help,”—P1, M, age unknown. However, one participant acknowledged it “comes down to, you know, a personal choice, you go to the doctors to see them because… when you’re like concerned and you want to get a second opinion from your GP,”—P25, F, 49 years old.

A few participants reported having negative experiences with their GP, which led to loss of trust with their GP, as “there was a bit of controversy there… It has taken me a long time after that doctor [for]… my trust and faith in the hospital, in the doctor [to come back]…”—P7, F, age unknown. Other participants reported there were times when participants did not feel heard by their healthcare provider “because [the GP] had already put things in place without listening… and I know he doesn’t advocate antibiotics, but this particular time moko [my grandchild] needed it,”—P9, F, 63 years old.

#### 2.2.3. Individual Factors

##### Illness Perceptions, Treatment Beliefs and Whaakaro [Thoughts]

Participant illness perceptions, treatment beliefs and whaakaro were important factors that emerged from the interviews. There was variability between participants in their beliefs about antibiotics. Some participants described being happy to receive antibiotics as they believed this would be the best or only treatment for them to get better, providing “a bit of a relief that I’m going to be fixed and feeling better and I can do what I want, you know,”—P19, M, age unknown. Participants stated that they believed getting antibiotics was a necessity, that “it is a must now… especially for your children. If your children need it, give it. It’s worse if they suffer,”—P5, F, age unknown. Participants discussed other positive treatment beliefs and emotions associated with getting antibiotics and the help that they believed they needed—including feeling “really happy, grateful, thankful. Yeah, I was so happy that someone had helped me,”—P1, M, age unknown. 

In contrast, participants also discussed that antibiotics were more harmful than useful, expressing concerns about the effect of antibiotics on the body: “[antibiotics are] more harmful than anything else. Nah, I just don’t like it at all,”—P6, F, age unknown. Some also believed that “if you take too much, you could damage your [body] system more,”—P26, F, 45 years old. There were concerns about adverse effects caused by antibiotic use that influenced participants’ ideas about antibiotic use. This led to avoidance of antibiotic use because of participants’ previous negative experiences, including side effects and misdiagnosis. In one example, the “doctors have prescribed my moko [kids]… with the wrong antibiotic and the side effects [occurred],”—P17, F, age unknown. In some cases, this prevented participants from accessing the antibiotics, “because whatever is in it gives me bad thrush… so it’s a waste of time coming here to get you [antibiotics] if I have to come back and get you [thrush medication],”—P6, F, age unknown. Other participants were adamant that they did not want antibiotics at all unless it was the last resort: “I try not to take antibiotics unless I really have to,”—P25, F, 49 years old.

##### Natural (Rongoā) vs. Western Medicine

Participants highlighted the importance of incorporating natural medicine as part of their healing process with western medicine. Participants perceived alternative healing as aligning with their spiritual and physical well-being: “I take alternate healing. Sometimes the wairua needs to be aligned as well and I always believe that because my father was a healer, so he did healing with his hands,”—P9, F, 63 years old. This was particularly important for some as they were raised with natural medicine, which has influenced participants’ perceptions of medication used today: “when I got brought up, my grandmother’s like, some of the medicines she just didn’t agree, so she would go out and pick these leaves and just like… yeah, and it worked,”—P16, F, age unknown. Many participants discussed preferring natural medicine such as rongoā, Māori medicine, over pharmaceuticals like antibiotics, perceiving that “rongoā is better than anything,”—P3, F, 20 years old, and that “rongoā would work better,”—P6, F, age unknown. In contrast, a few participants reported not having exposure to rongoā or not having exposure to the information regarding rongoā, stating that they have “never used it, but …have recommended it because [they] have been told about it … but… never used it … because [they] don’t know where [they] can get anything like that,”—P27, F, 44 years old. 

#### 2.2.4. Potential Solutions to Improve Antibiotic Use

Almost all participants provided suggestions to improve antibiotic use. Suggestions centred on better communication about antibiotics, with a focus on improving people’s knowledge about antibiotics. Participants felt that GPs “should give out more info[rmation] to our people [Māori] or to whoever, give out more information and explain to them that antibiotics that they’re giving to them,”—P6, F, age unknown. There were suggestions about disseminating information using pamphlets and presentations with accessible health information as an engaging way to deliver messages about antibiotics, to “…have more info[rmation] about [antibiotics], even with pamphlets, in the health thing, and all that. They got all the you need to stop smoking—you need to do this, medication wise. They need pamphlets for the actual medication they’re giving out… Yeah, sort of do like a presentation thing… on the marae,”—P26, F, 45 years. Other participants reinforced the importance of considering how the message was delivered, for example, some participants suggested the marae as a good venue to share information when learning about antibiotics as the “marae has always been a place, as a kid, that we always learnt a lot on our maraes” and “it’s an open place for kōrero, for everything,”—P25, F, 49 years old. Having conversations face-to-face with health professionals was thought to important to increase antibiotic knowledge: “…a lot of verbal info[rmation], like I hate reading, like give me that, I’ll go home and read it but I hate reading… Sitting down and having a conversation about it… and you telling me directly the good and the bad… Well kanohi ki te kanohi … is the best, we need to sit down and talk. Never mind about all this paper trail thingy,”—P23, F, 66, Waikato/Ngāpuhi. This was supported by the view from some participants who recognized that engagement with their GP was an important aspect in being able to have good knowledge in relation to antibiotics, which can help address any illness or treatment misconceptions. Some believed that their GP needed to engage better with them and their whānau to improve understanding about antibiotics, as “…health literacy is important. I think that… they need to know more, and health professionals need to know how to engage and educate you. Know the consumer, whānau, so that they have an understanding of what antibiotics actually do,”—P28, F, 28 years old.

Participants emphasised the importance of cultural support and involving Te Ao Māori into learning about antibiotics, for example having “more cultural support to engage people with antibiotics… Understanding the medicines and the interactions. Using Māori Reo and Tikanga as way to help,”—P5, F, age unknown. A few participants highlighted that this is a ‘Pākeha world’ involving antibiotics, and suggested using Te Ao Māori to help patients understand, as “…you can’t put a group of Māori in a room of Pākeha who are assessing that kōrero… So, it’s someone who can view both worlds and yeah, I just reckon that’s exactly what I think because you’ve got to learn both worlds to understand them,”—P28, F, 29 years old. A few participants also commented on prevention of infectious disease to avoid using antibiotics in the first place, that the focus of education should be “more healthy living and that can avoid you from having to come on to see the GP [for infections] if we can live a more healthier life,”—P24, M, 49 years old.

## 3. Discussion

This is the first study to explore Māori beliefs about antibiotics and their use for cold or flu symptoms. We adopted a qualitative research approach to allow exploration of the experiences and perceptions of Māori in-depth. We identified systemic, social and individual factors that influenced antibiotic use for Māori. Personal beliefs were influenced by the wider health systems and infrastructures within NZ, which also impacted on the social issues that people experienced. Most participants appeared to have low expectations for antibiotic treatment for cold and flu symptoms; many expressed that they did not want to have unnecessary antibiotics or reported experiencing adverse effects with previous treatment.

From the themes identified from this study, there are several key factors that could potentially be modifiable or targeted in an intervention to improve antibiotic use. First, knowledge relating to infections (viral or bacterial) and their management (symptomatic care or antibiotics) appeared to be lacking in this participant group. Participants described seeking information or clarification from other sources immediately after their GP consultation, using Google or whānau, which reflects a clear need for better, more accessible information in this population. As provision of information in way that is tailored to an individual’s health literacy level reflects the response of the health service or provider, and not simply the knowledge or capacity of the person seeking care, these findings suggest that health professionals need to develop and implement more effective resources for Māori who present with infections that may or may not require antibiotics (8, 10). Participants described a hierarchy during the consultation with the doctor, where the doctor’s advice was seen as an important ‘direction’ even if they did not understand the information about antibiotics. People felt that it was not safe to ask questions or question treatment decisions. Unfortunately for Māori in NZ, there is both under-prescribing of necessary treatments and over-prescribing of inappropriate medications [21], creating and sustaining health inequities [4,5]. There is potential for ‘good prescribing’ when patients and their prescribers discuss the reasons for the medicine, including risks and benefits [1,17,18]. Our findings suggest a need to address the relationship between health providers and Māori ‘10,21].

We identified issues of access and poverty that play a big factor in healthcare. Any relationship between the participants and their doctors has potential to be hindered by access and poverty issues. Participants described a cycle of Māori being unable to access care due to poverty and lack of transport, health professionals not seeing or understanding these issues and behaving in ways that caused misinformation or mistrust—for example, communicating information in a way that participants felt like they had not been listened to, and people’s beliefs about antibiotics being further undermined. This cycle needs to be broken. Any campaigns focusing on the education in Māori communities need to be done in a culturally safe way [22]. Programs that educate health professionals about appropriate antibiotic use have been shown to be effective in reducing unnecessary prescribing of antibiotics [23]; attention to developing similar programs for prescribers working with Māori communities may be useful. 

Participants highlighted a key point about their preference not to take antibiotics or at least for health providers to consider incorporating traditional medicine or alternative Māori traditional remedies as part of their treatment plan. This was reinforced by participants’ negative experiences and past adverse effects with antibiotics, which impacted their preference to seek alternative remedies (rongoā). Ancestral knowledge was lost because of colonisation with unequal treatment of Māori within societal institutions displacing many cultural traditions such as tohunga [24]. The Tohunga Suppression Act in 1907 removed the use of rongoā within Māori society, as Pākeha believed it to be witchcraft [24]. As a result, western medicine, including antibiotics, was introduced and as with most ‘health’ interventions in NZ, inequities in infection rates, management (including antibiotic prescription) and outcomes followed [25]. Participants highlighted the need for the NZ healthcare system to respond and acknowledge Te Ao Māori as part of care, recognising the impact of colonisation in relation to the access barriers they faced with health care and health information [25] leading to differential access to material resources such as income, resulting in lower socio-economic status [18]. This can lead to a loss of financial opportunity to access healthcare, which can lead to barriers to seeking necessary health information and exacerbate poor health literacy. Although complex, the role of colonisation needs to be addressed in future antimicrobial stewardship interventions and in ongoing research [26].

These findings have important implications for antimicrobial stewardship initiatives in NZ. Participants suggested solutions to support and promote optimal antibiotic use in Māori. Developing a framework that addresses the downstream determinants of health such as information and patient experience with antibiotics would be key. Targeting this area would improve health knowledge and work to improve health literacy [21]. It would incorporate solutions as described in our findings, using a Māori worldview to engage Māori communities. 

There is a tension that exists with antibacterial use and the inequities that are seen in both infection and prescribing rates between Māori and non-Māori. Care is required to balance appropriate antibiotic use with the risks of resistance that can arise from overuse. Nationally, the drive to reduce antibiotic use to reduce the risk for antimicrobial resistance in NZ is countered within a context of Māori missing out on appropriate and necessary antibiotics relative to health need [21,27]. This study adds much to understanding and addressing this unique issue.

A potential intervention could be one informed by constitutional reform, for example, one that challenges colonisation and acknowledges the power imbalance between Māori and Pākeha [28]. Constitutional reform would focus on Māori self-determination and tino rangatiratanga being enforced on a governmental level [29]. The recently announced Health and Disability System Reforms, including the establishment of a Maori Health Authority, is a step in the right direction, in which Māori are provided the necessary opportunity to make decisions for Māori by Māori, with a focus on creating equitable outcomes for Māori health, including appropriate antibiotic use, and reducing infection rates. 

There were several strengths to the research. Interviews used in this research provided an opportunity for open discussion with Māori in a comfortable setting. The study focused on Māori participants and an area of healthcare that has not been studied in NZ before, filling a knowledge gap. The qualitative nature of the research allowed a wide variety of ideas, opinions and beliefs to be expressed and provided a crucial starting point to address inequities regarding antibiotic prescribing in NZ. Limitations to the study included sampling and methodological issues. Whilst a sample size of 30 is generally sufficient in qualitative research to ensure data richness and to reach data saturation [30], a larger sample recruited via different settings may identify different themes. The population studied also came from only one clinic, hence the results may not be generalizable to other regions or settings. However, the data from this study can be used to inform future studies such as additional in-depth qualitative studies in other regions or future quantitative research, for example, a national survey exploring beliefs about antibiotics and AMR. Further research using kaupapa Māori methodology is also recommended to build on our findings [31]. Our study also did not explore the perceptions and experiences of non-Māori. As such, we do not know how our findings directly compare to the general population; a study comparing the perspectives of Māori and non-Māori may now be warranted. Steps to incorporate more Te Reo Māori into the structure of interviewing, include whakapapa as a means of connection to the participants and have interviews in safe cultural spaces, such as on a marae, may also be useful. 

## 4. Materials and Methods

### 4.1. Patient Recruitment

Participants were recruited from a general practice with a high Māori patient population (Papakura Marae Health Clinic). Papakura Marae is located in Papkura, a south-eastern suburb of Tāmaki Makaurau, approximately 32 kilometres south of the Auckland central business district. The Papakura Local Board area has a population of 45,633, with 24% of residents being 15 years and under (compared to 20.9% across Auckland) and the highest number of Māori (28.1%) within its Local Board boundary (compared to 10.7% across Auckland) [32]. A sample of 30 adults aged 18 years and over, who presented with symptoms of an upper respiratory tract infection (cold or flu symptoms) to Papakura Marae Health Clinic, were invited to participate in the study. This target sample size was selected as a sample of 20 has been recommended to ensure data richness and thematic saturation in qualitative research [30]. Participants were identified as suitable for inclusion in the study by doctors, nurses and pharmacists working at the Marae, who ensured the participants were able to participate in the interview process and provide informed consent. No formal exclusion criteria were applied.

Patients identified as suitable for inclusion and who consented to find out more information about the study were then introduced by the health worker to the interviewer (KH). The study was explained to them by the interviewer and when patients agreed to participate, verbal and written informed consent was obtained from all participants prior to the interview. All participants had sufficient time to discuss with whānau (family, see Appendix A for Te Reo definitions) or others about the study, if desired. Interviews were conducted onsite. The study was reviewed and approved by the New Zealand Health and Disability Ethics Committee (19/STH/203).

### 4.2. Interview Process

The interview process consisted of a 20- to 40-minute discussion using a semi-structured guide (Appendix A) developed by the research team based on their research experience and on the published literature [16,17]. The investigator team reviewed the interview guide and pilot tested the wording of the questions for clarity and flow. Participants could participate either in a one-on-one interview with the researcher (KH), or in a focus group with up to 3 other participants, depending on participant preference and availability. A combination of both interviews and focus groups were used to increase participant response and enhance data richness [33]. Each participant was allocated a code to allow de-identification. All participants were interviewed on-site at the GP clinic in a private consultation room. Refreshments were available for the participants during the interview process. After the interview participants were given a koha (gift/donation) of a NZD$20 grocery voucher. Interviews were audio recorded, but in situations in which participants did not wish to be audio-recorded, written notes were made as an alternative.

### 4.3. Data Analysis

Audio recordings were transcribed verbatim, but any potentially identifying information was removed. The data collected through the audio transcription and field notes were then analysed by manual coding using NVivo 12 software, using an inductive thematic analysis approach. Data were coded and common themes were derived iteratively and described, focusing on key themes and subthemes that reflected participant views and experiences with antibiotics and wider factors that may influence antibiotic use. Transcripts were read at least twice by the interviewer (KH) to ensure consistency of coding. All themes were second-coded and checked by another member of the research team (AC). 

## 5. Conclusions

There are many factors influencing antibiotic use within Māori communities, at a system, societal and individual level. Importantly, the study highlighted key socioeconomic determinants for Māori that impact their use of antibiotics, and their experiences of the health system, including what to do when having an infection and how to receive appropriate treatment. These socio-economic inequities for Maori can be traced to historical and contemporary colonisation and attempts to address these upstream factors are likely to require political transformation to a system in which Maori leadership is supported. Considerations need to be made to consider Te Ao Māori in healthcare when prescribing antibiotics, and as lack of recognition of this can lead to negative downstream effects in health for Māori, such as increase in infections and substandard antibiotic use. Māori face reduced consultation times, leading to less rapport with health professional, decreased health literacy regarding antibiotic use and an increase in infection rates.

## Figures and Tables

**Figure 1 antibiotics-11-00714-f001:**
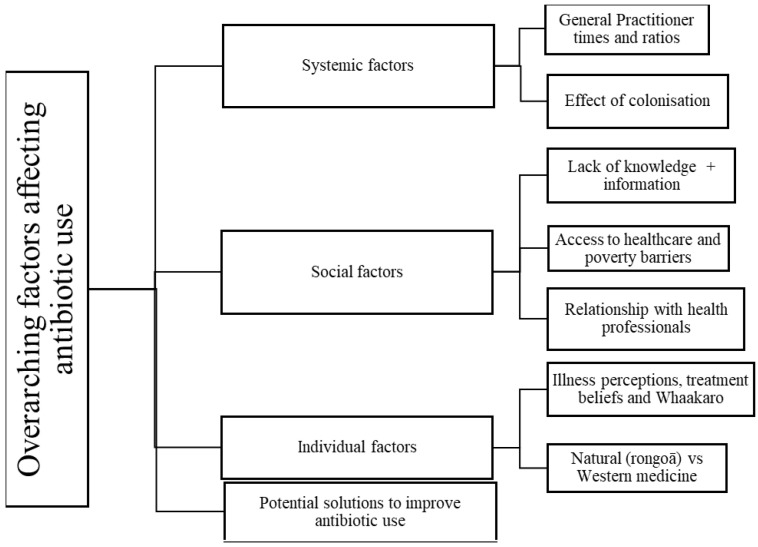
Themes regarding antibiotic use derived from interviews with 30 participants.

## Data Availability

Not applicable; our ethics approvals do not allow sharing of interview data collected.

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
