# Peer review of "Māori Experiences and Beliefs about Antibiotics and Antimicrobial Resistance for Acute Upper Respiratory Tract Symptoms: A Qualitative Study"

_antibiotics, 2022, doi:10.3390/antibiotics11060714_

Round 1

Reviewer 1 Report

Manuscript, entitled "Māori experiences and beliefs about antibiotics and antimicrobial resistance", reports the findings of a quantitative study focused on experiences and belief about antimicrobial therapy and AMR. The manuscript was well written and interesting. However, the manuscript  was more like a narrative report, not a scientific study. 

Major concern:

  1. The study title should be "Māori experiences and beliefs about antibiotics and antimicrobial resistance on acute upper respiratory tract symptoms".
  2. The abstract did not reflect all important findings of the study. Only outline of the study was elaborated in the abstract. 
  3. The background section should also include previous studies focusing on perception/belief of prescribers/patients about antibiotic use for treatment of acute URI (PLoS One. 2021 Feb 11;16(2):e0246782, BMC Fam Pract. 2019 Feb 14;20(1):27)
  4. The result section is more like a narrative report, not a scientific report. It would be better if the author could include a table included all key findings and a total number of participants who answer the same. For example - number (%) of participants who reported using the internet to find an answer about antibiotic use.
  5. The method section should include more information regarding the study settings. General demographic data of Maori community).  How did they enroll all participants? Did they provide any incentive? Did the study team test the study questions with a few pilot subjects before performing an interview?
  6. The conclusion section should mention about future studies. Data from this study can be used for preparation of semi-quantitative study to explore this issue.

Reviewer 2 Report

The concept /aim of the study is relevant to provide some baseline data on driving forces of antibiotic use among Maori people. Objectives are clear, well written paper which is easy to understand. study design is good and reproducible.
However, these my few suggestions for consideration to authors;

Key words in Abstract too many words, should be revised and simplified  to i.e : 
Māori, antibiotics, prescribing, use, beliefs, experiences, antimicrobial resistance.

Introduction

Page 2 of 11 line 74: Suggest deletion of '(colds and flu)' from the objective. It can be explained in the introduction, methods etc

Results

Page 2 Line 77-78": ....interviewed 1:1... this aspect of sentence not clear. DO they mean participants were interviewed one-on-one?

Results

Page 9 of 11 Line 408:

insert/add to limitations:....small sample size

Reviewer 3 Report

Very interesting idea behind the study, however I have some issues to raise:

Introduction

Can health inequities for Maori be explained in greater detail?

Also, could you give a background on Maori culture, i.e. do they undergo general education etc.

Methods

Have all participants provided a written informed consent?

there was no sample size calculation? This should be discussed as a limitation to the study.

Also, I have concerns regarding the variability of interviews and focus groups.

I highly doubt this is a cohort, please reconsider.

This study would have been more interesting would there have been a control group of non-Maori New Zealanders

Results

I don't think you adequately addressed the question of antibiotic use in Maori, as you did not investigate their practices and common practices that may add to resistance i.e. use of leftover antibiotics. Also you did not provide information whether traditional medicine may interfere with antibiotics i.e.

Results section in general is very difficult to read.

Discussion section

Overall, I do not agree traditional medicine should be observed as evidence based medicine, regardless of culture. Education of Maori would be a better approach then accepting 'witchcraft' as treatment.

Round 2

Reviewer 1 Report

-

Author Response

We have reviewed the English language and amended according to the Editor comments